# Fractional re-distribution among cell motility states during ageing

Jude M. Phillip [1,2✉], Nahuel Zamponi[3], Madonna P. Phillip[2], Jena Daya[2], Shaun McGovern[2],
Wadsworth Williams [2], Katherine Tschudi[2], Hasini Jayatilaka[4], Pei-Hsun Wu [2], Jeremy Walston [5] &
Denis Wirtz [2,6]

Ageing in humans is associated with the decreased capacity to regulate cell physiology. Cellular properties, such as cell morphology and mechanics, encode ageing information, and can therefore be used as robust biomarkers of ageing. Using a panel of dermal fibroblasts derived from healthy donors spanning a wide age range, we observe an age-associated decrease in cell motility. By taking advantage of the single-cell nature of our motility data, we classified cells based on spatial and activity patterns to define age-dependent motility states. We show that the age-dependent decrease in cell motility is not due to the reduced motility of all cells, but results from the fractional re-distribution among motility states. These findings highlight an important feature of ageing cells characterized by a reduction of cellular heterogeneity in older adults relative to post-adolescent/adults. Furthermore, these results point to a mechanistic framework of ageing, with potential applications in deciphering emergent ageing phenotypes and biomarker development.

[1] Departments of Biomedical Engineering, Johns Hopkins University, Baltimore, MD 21218, USA. [2] Department of Chemical and Biomolecular Engineering, Institute for NanoBiotechnology (INBT), Johns Hopkins University, Baltimore, MD 21218, USA. [3] Department of Medicine, Hematology and Oncology Division, Weill Cornell Medicine, New York, NY 10065, USA. [4] AtlasXomics Inc., New Haven, CT 06511, USA. [5] Department of Medicine, Division of Geriatric Medicine and Gerontology, Johns Hopkins University School of Medicine, Baltimore, MD 21224, USA. [6] Departments of Oncology and Pathology, Sidney Kimmel Cancer Center, Johns Hopkins University School of Medicine, Baltimore, MD 21287, USA. ✉email: jphillip@jhu.edu

Ageing can be defined as the accumulation of dysfunctions with the passage of time that limits the ability of organisms, organs, and tissues to absorb and rebound after perturbations and stressors[1–3]. In humans, normal ageing is associated with diverse physiological changes that influence the magnitude, and rates of progressive decline among individuals. These include the decreased abundance and activity of circulating cytotoxic immune cells, slower gait speed, and declines in cardiorespiratory fitness[4–8]. Furthermore, the high variability among individuals suggests that there is no uniform ageing phenotype. A growing body of evidence shows that the interactions of intrinsic and extrinsic factors such as molecular states[9–12] (e.g., epigenomic) together with environmental factors and macroscopic stressors[13–15] (e.g., social determinants and disparities) contribute to the rates of ageing in individuals[16]. However, it remains unclear how the underlying molecular states of an individual relate to their clinical outlook at the organ/tissue level in the context of ageing. We postulate that studying age-associated changes at the intermediate length scale of cells themselves—between the larger length scale of organs and tissues and the smaller length scales of molecules—may provide the missing link to understand the inter-relation of these ageing scales[3,17].

As integrators of molecular signals, cells offer a sensitive meso-scale view of ageing, with cellular dysfunctions likely occurring prior to the manifestation of age-related disorders and diseases at the clinical level. Populations of cells typically display dynamic and heterogeneous phenotypes in the context of health and disease[18,19]. In a previous study, we demonstrated that ensemble functional biophysical properties of cells, such as cytoplasmic stiffness, force generation, and morphology, which typically capture time-independent (i.e., snapshot) cellular phenotypes, encode essential ageing information that can be used as robust biomarkers of ageing in healthy individuals[17]. However, it is still unclear how this ageing information is encoded, and its potential role in developing innovative approaches for precision health. We postulate that the tracking and analysis of dynamic ageing phenotypes at the cellular level could provide a unique perspective, and offer mechanistic insights into the ageing process[20]. Furthermore, this approach to study dynamic cell properties may provide more information compared to snapshots of cell phenotypes obtained from fixed cells.

In this study, we analyze single-cell motility patterns of primary dermal fibroblasts obtained from healthy donors spanning an age range from 2 to 92 years old. Using a combination of bulk and single-cell analysis tools, we show that cells can be classified into various motility states based on spatial and activity (i.e., consistent versus sporadic) patterns. We then demonstrate that the age-associated decrease in overall cell motility was linked to the fractional re-distribution of cells among the identified motility states, with a significant decrease in cellular heterogeneity for older adults (>65 years) relative to post-adolescent adults (29–65 years).

## Results

### Global decrease in bulk cell motility with increasing age.

Coordinated cell movements are essential for the development of tissues and organs, in homeostasis and disease[3,17]. As cells move, there is an intricate coordination of biophysical and biomolecular programs that change with age, some of which involve the modulation of cellular biomechanics, adhesion and regulated dynamics of the cytoskeleton within cells[21,22]. To elucidate possible age-related changes in cell motility patterns, we procured a panel of primary dermal fibroblasts from healthy donors spanning an age range, from 2 to 92 years old (Supplementary Data 1). Using time-lapse microscopy, we recorded and tracked the spontaneous movements of these cells seeded on type-I collagen-coated substrates (see "Methods" section). Analyzing cell trajectories (Supplementary Data 2), we computed averaged mean-squared displacements (MSDs) of cells and found trends towards an age-dependent decrease in overall cell motility[17,23] (Fig. 1A, B). In particular, the values of mean-squared displacements (MSDs) evaluated at time lags of 6 and 60 min (MSD6 and MSD60) and corresponding average speeds (SP6, SP60) decreased steadily with age (Supplementary Fig. 1A–J). These time lags of 6 and 60 min were chosen because they correspond to time scales shorter and longer than the average persistence time of cell motility across all ages (Supplementary Fig. 1A–D).

Building on these findings, we then asked whether cells taken from young donors displayed distinct spatial motility patterns compared to cells derived from older adults. Analyzing the motility data based on the recently introduced anisotropic persistent random walk (APRW)[24,25], a framework that recognizes that cell trajectories do not always follow random walks even on 2-dimensional substrates, we first assessed the similarity of cell movements per unit time, given by the magnitude of the autocorrelation of cell velocities (see "Methods" section). We observed a faster decay in the autocorrelation function of successive migratory steps with increasing age (Fig. 1C), which suggests shorter persistence times, or more frequent changes in the direction and velocity of cells with increasing age. We then asked whether this bulk decrease in motility was accompanied by a bias in the spatial polarity of cell movements, or a similar likelihood of movements in all directions. Quantifying the angular velocity profiles of cells, we found that cells from young donors exhibited an ellipsoidal profile of angular velocities and a tendency towards a circular profile for cells from older adults (Fig. 1D). This indicates a loss in spatial persistence and directionality of cell trajectories with increasing age. Together, these results indicate that dermal fibroblasts show a loss in both temporal and spatial ensemble persistence with increasing age, with cells from older adults displacing less with frequent changes in their movement direction.

To systematically define bulk age-dependent motility patterns, we computed the Pearson correlation coefficients for ten parameters that describe age-dependent spatial movement patterns of cells (see "Methods" section). These parameters include the magnitudes of cellular displacements and speeds (MSD6, MSD60, SP6, and SP60), the total diffusivity and diffusivities along primary and secondary axes of migration (Dtot, Dp, and Dnp), the persistence times along the primary and secondary axes of migration (Pp, Pnp), and the spatial persistence/anisotropy (ϕ). This analysis showed negative age-associated correlations for all motility parameters (Fig. 1E, F and Supplementary Fig. 1K, L, Supplementary Data 3), further highlighting the notion of overall decreased cell motility with increasing age (Supplementary Fig. 1A–J).

Given the significant age-associated changes in cell motility, we asked whether the motility patterns of individual cells could provide insights that are not fully appreciated from the above bulk quantification. Plotting the $x$–$y$ trajectories for all cells on the same length scale, we qualitatively observed the aforementioned global decrease in cell displacements based on the origin-centered footprint of cell trajectories with increasing age (Fig. 1G, top panels). However, closely examining the movement patterns of individual cells, we observed extensive cell-to-cell variations and the presence of cells having both motile and non-motile patterns from the same donor (Fig. 1G, bottom rows).

### Age-dependent decrease in cell motility corresponds to a redistribution among spatial clusters. Prompted by the

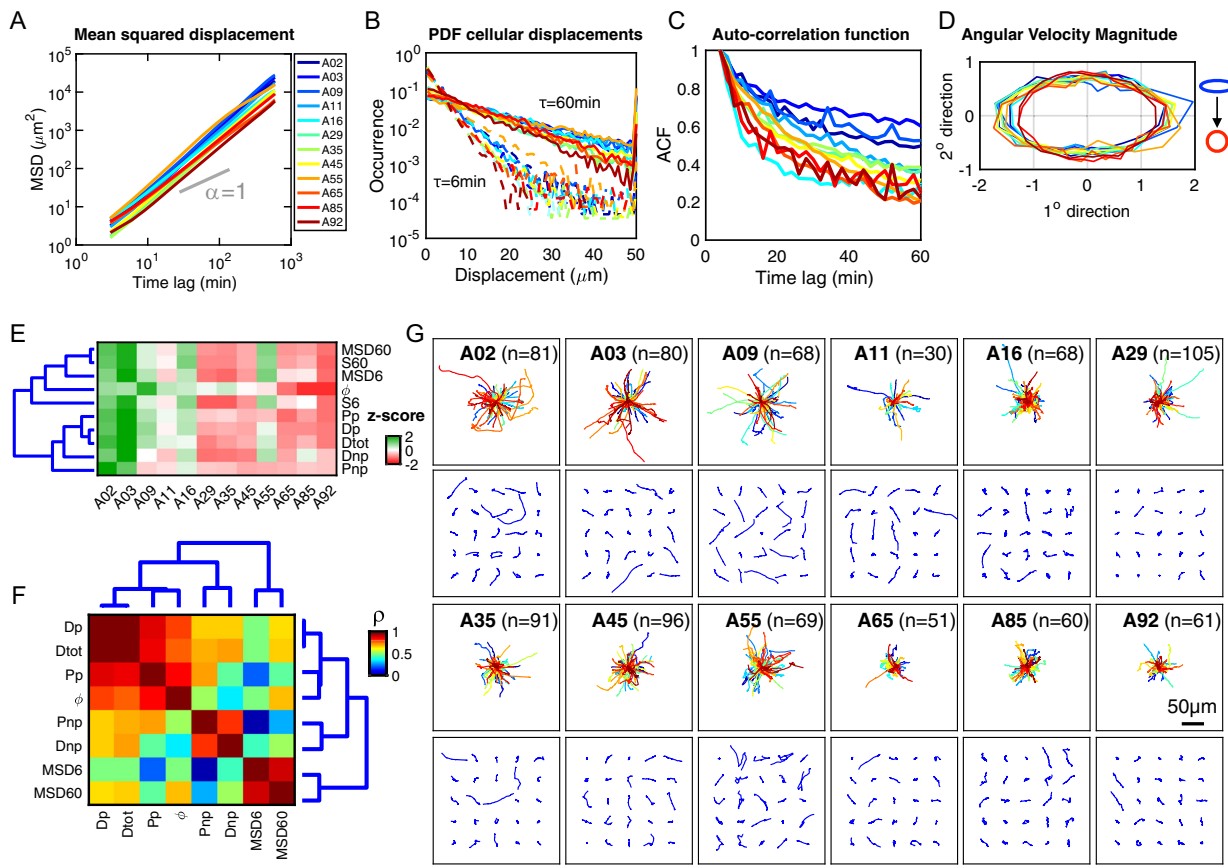

**Fig. 1 Global decrease in ensemble cell motility with increasing age. A** Age-dependent mean-squared displacements (MSD) of dermal fibroblasts with color code indicating the age of the donor (blue-to-red), an exponent alpha of 1 denotes pure diffusion. **B** Age-dependent probability functions of cell displacements at time lags equal to 6 min (dashed lines) and 60 min (solid lines). **C** Auto-correlation function of velocities (ACF) measured at 3 min time lags. **D** Angular velocity magnitudes as a function of the donor age, "circular" denotes similar likelihood for all angles and "ellipsoidal" indicating a polarization of cell movements along primary axis of migration. Color coding in **B** also applies to **C**, **D**. **E** Heatmap showing the magnitude of the correlation between motility parameters ($z$-score normalized) and age. Dendrogram branches indicate unsupervised hierarchical clustering with ward linkages of the cityblock distances of parameters. Red-to-green signifies low-to-high Pearson correlation coefficient. These parameters include total diffusion (Dtot), diffusion along the primary and secondary axes of cell movements (Dp and Dnp), measure of the spatial persistence—anisotropy of the cell movements ($\phi$), persistence time of cell motion along the primary and secondary axes of cell movements (Pp and Pnp), average cell speed at time lags of 6 and 60 min (S6 and S60), and the MSD measured at time lags of 6 and 60 min (MSD6 and MSD60). **F** Heatmap showing the magnitude of cross correlations of motility parameters across all cells for all ages, range of Pearson correlation coefficients = 0.39–0.94. **G** Cell migration patterns and trajectories for primary dermal fibroblasts collected from healthy donors with ages spanning 2–92 years. Top panels show origin-centered trajectories for all cells per age; bottom panels show a grid of $x$–$y$ trajectories for 25 randomly selected cells per age. The number of tracked cells per sample is indicated in the upper left corner of the plot.

magnitude of the observed cell-to-cell variations in cell movement trajectories (Fig. 1G), we hypothesized that the age-dependent decreases in cell motility was not due to decreased cell movements in all cells, but results from an age-dependent redistribution of the proportions of motile versus non-motile cells. Pooling the cell trajectories across all ages, we first log-normalized the motility parameters defined above, then computed the $z$-scores per parameter to allow direct comparison across the same numerical scale (Supplementary Fig. 2 and see "Methods" section). Using unsupervised hierarchical clustering (see "Methods" section), we determined inherent cell-based and parameter-based groupings using the "City block" distances along the axis of maximum variation (ward linkages). We identified eight spatial clusters (Pn) based on groups of cells having similar magnitudes of displacements/diffusivity, and spatio-temporal persistence (Fig. 2A). In addition, we identified three clusters based on the similarity of trends among motility parameters for all ages (Supplementary Fig. 3A, B), primarily describing trends related to

magnitude of displacements-G1 (Dp, Dtot, MSD6, and MSD60), persistence/directionality in primary axis of movement-G2 (Pp, $\phi$) and non-persistence-G3 (Pnp, Dnp) (Fig. 2A, B).

To understand the differences among cell-clusters and decipher what each cluster represented, we plotted the trajectories of individual cells within each cluster. Visual inspection indicated distinct patterns of cell movements in each cluster (Fig. 2C). For instance, cluster 3 (P3) corresponded to cells having a high degree of persistence and diffusivity, whereas cluster 8 (P8) comprised of non-motile cells characterized by small displacements, low diffusivity and low persistence (Supplementary Fig. 4).

Continuing to address our hypothesis of fractional redistribution of cells, we plotted two-dimensional t-stochastic neighbor embedding (t-SNE) maps for all cells (each dot represents a single cell). Coloring them based on the eight spatial clusters determined using the hierarchical clustering, we observed a confirmation of segregated groups of cells (Fig. 2D). Using this same t-SNE layout, we then painted each cell according to their

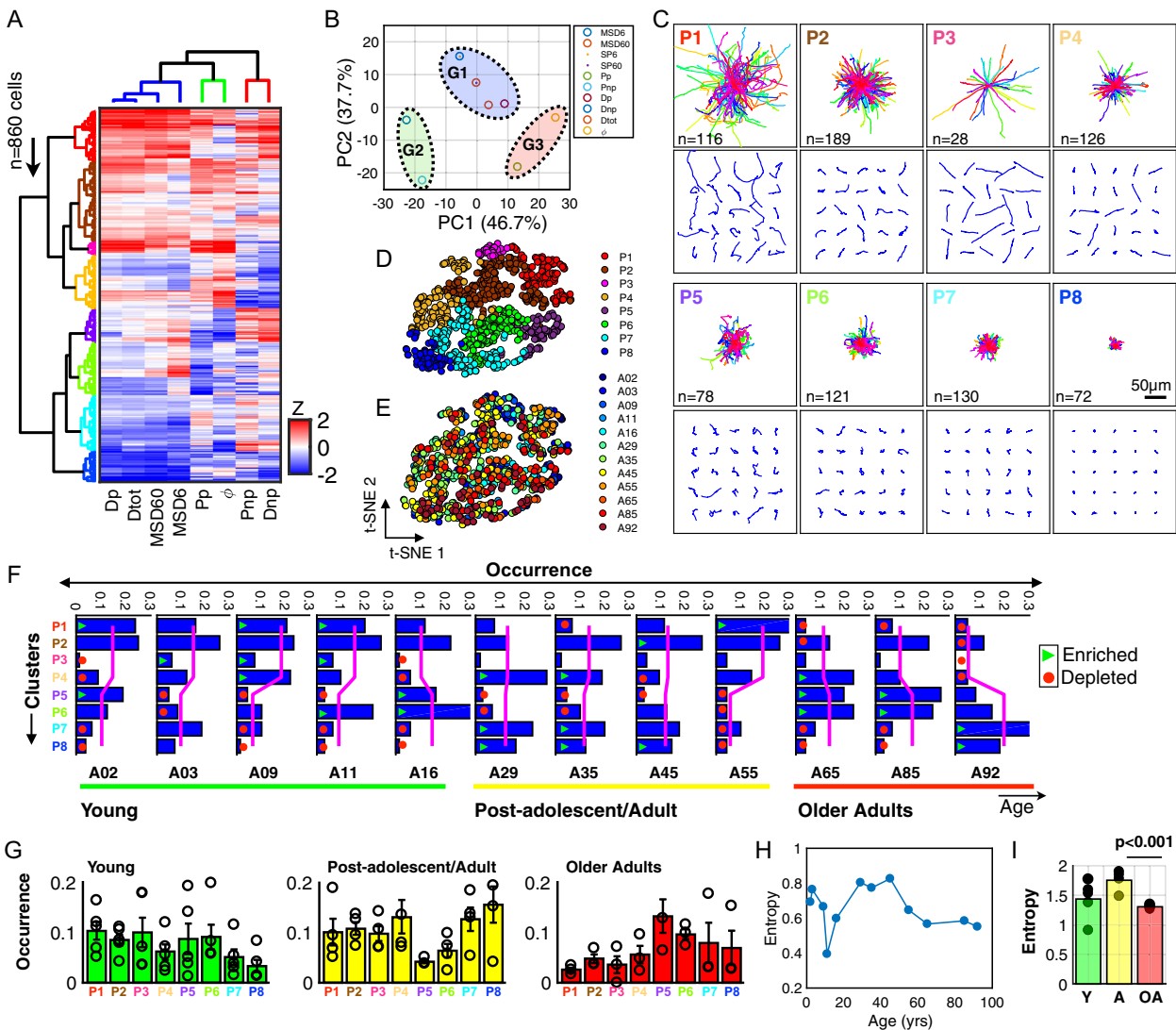

**Fig. 2 Age-associated spatial motility patterns at the single-cell level. A** Heatmap showing eight motility parameters measured at the single-cell level across all ages ($n = 860$ cells); each row represents a single cell and each column a motility parameter. Dendrogram branches represent hierarchical clustering using the city block distances with ward linkage along axes of maximum variation. Data shows the delineation of single cells into eight spatial motility clusters, with the column-clusters ($n = 3$ clusters) describing the degree of cellular displacements, and spatio-temporal persistence. **B** Principal component analysis (PCA) of the eight motility parameters showing the delineation into three groups, G1-high displacement cells, G2-persistent, and G3-non-persistent. **C** $x$–$y$ trajectories based on the eight clusters that delineate patterns of cellular movements. Top panels show the origin-centered trajectories for all cells within clusters; bottom panels show $x$–$y$ trajectories for 25 randomly selected cells per cluster, with the number of cells per cluster indicated in each plot, scale bar = 50 μm. **D** Two-dimensional t-SNE visualization for the eight color-coded spatial clusters. **E** Age-painted t-SNE plot showing the distribution of ages per spatial motility cluster. **F** Frequency distributions indicating the fractional composition of cells per cluster as a function of age. Magenta trend lines represent the average fraction per primary branch of the dendrogram tree. The first bifurcation delineates four clusters each. Trend lines show a progressive transition in the abundance of cells across primary branches with increasing age, green triangles denote significantly enriched and red circles denote significantly depleted ($p$-value < 0.05). **G** Frequency distributions showing the average fractional abundances of cells per cluster separated into three groups, data points per age shown as black points and error bars indicate the standard deviation; young, post-adolescent/adult, and older adults. **H** Magnitude of the Shannon entropy for each individual sample with increasing age. **I** Average Shannon entropy based on these three age groups, showing a decrease in heterogeneity of motility from post-adolescent/adults to older adults ($p < 0.001$).

respective ages to determine whether certain clusters were defined by cells from a particular age (Supplementary Fig. 5). Interestingly, we found that cells from the same donor were intermittently distributed among all eight clusters (Fig. 2E). To appreciate the fractional abundances of cells within each spatial cluster (Pn), we plotted the frequency distributions for all cells per donor (Fig. 2F and Supplementary Data 4), which revealed progressive age-associated changes in the abundance of cells within various clusters. Specifically, cells from young donors tended to favor motility patterns described by P1–P4, while cells

from older adults favored phenotypes described by P5–P8 (magenta lines, Fig. 2F).

Together, results indicate an inherent polarization of the motility patterns based on spatial clusters exhibited by cells derived from young and older adults. In addition, cells derived from post-adolescent/adults exhibited flattening of the average abundance of cells from the two major groups of clusters (magenta lines), suggesting age-associated changes in cellular heterogeneity. However, even though we observed a general trend regarding the age-dependent polarity in abundances, individual

donors exhibited unique cluster-profiles (i.e., abundances for each spatial cluster) (Fig. 2F).

To quantify this age-associated heterogeneity, we grouped donors into three age groups based on categories defined in the literature to be associated with various clinical ageing characteristics[26,27]; young (A02, A03, A09, A11, and A16), post-adolescent/adults (A29, A35, A45, and A55) and older adults (A65, A85, and A92), then computed the Shannon entropy (S) within each age group, defined as[28] (Fig. 2G):

$$S = -\sum_{i}^{N} p_i \cdot \ln(p_i),$$

Here, $p_i = n_i/N$ is the proportion of cells belonging to each spatial cluster Pn, where $n_i$ is the number of cells within spatial cluster $i$, and $N$ is the total number of cells within that spatial cluster. The Shannon entropy measures the degree of uncertainty/ disorder within a distribution. In our case, the entropy is used as a surrogate for the intrinsic heterogeneity for a population of cells based on the abundance of cells within each of the defined spatial clusters. Here, the more uniform the distribution (i.e., similar abundance/flatter) the greater the entropy (Fig. 2H–I).

In sum, our results indicate that individual cells can be classified based on spatial patterns of their movements, with the magnitude of the average age-associated motility being approximated as the sum of weighted averages among spatial clusters per age (Supplementary Fig. 6), with a reduced cellular heterogeneity for older adult donors (>65 years) relative to post-adolescent adults and young donors.

**Cellular activity helps to describe age-dependent cell motility.** We next asked whether cells exhibited age-dependent differences in their activity patterns in efforts to better capture the intrinsic bursty dynamics based on the cell movement profiles[29]. This analysis provides insight into the fraction of time cells spend moving versus at rest, and whether cells moved consistently or sporadically. To determine the activity of individual cells, we converted each two-dimensional $x$–$y$ trajectory (Fig. 3A) into a one-dimensional displacement profile (Fig. 3B, see "Methods" section). We then computed the activity profile of each cell, normalized based on the $z$-score, then determined the magnitude (size of peaks) and frequency (number of peaks) of cellular movements. Pooling the activity profiles for all cells across all ages, we utilized unsupervised hierarchical clustering based on the

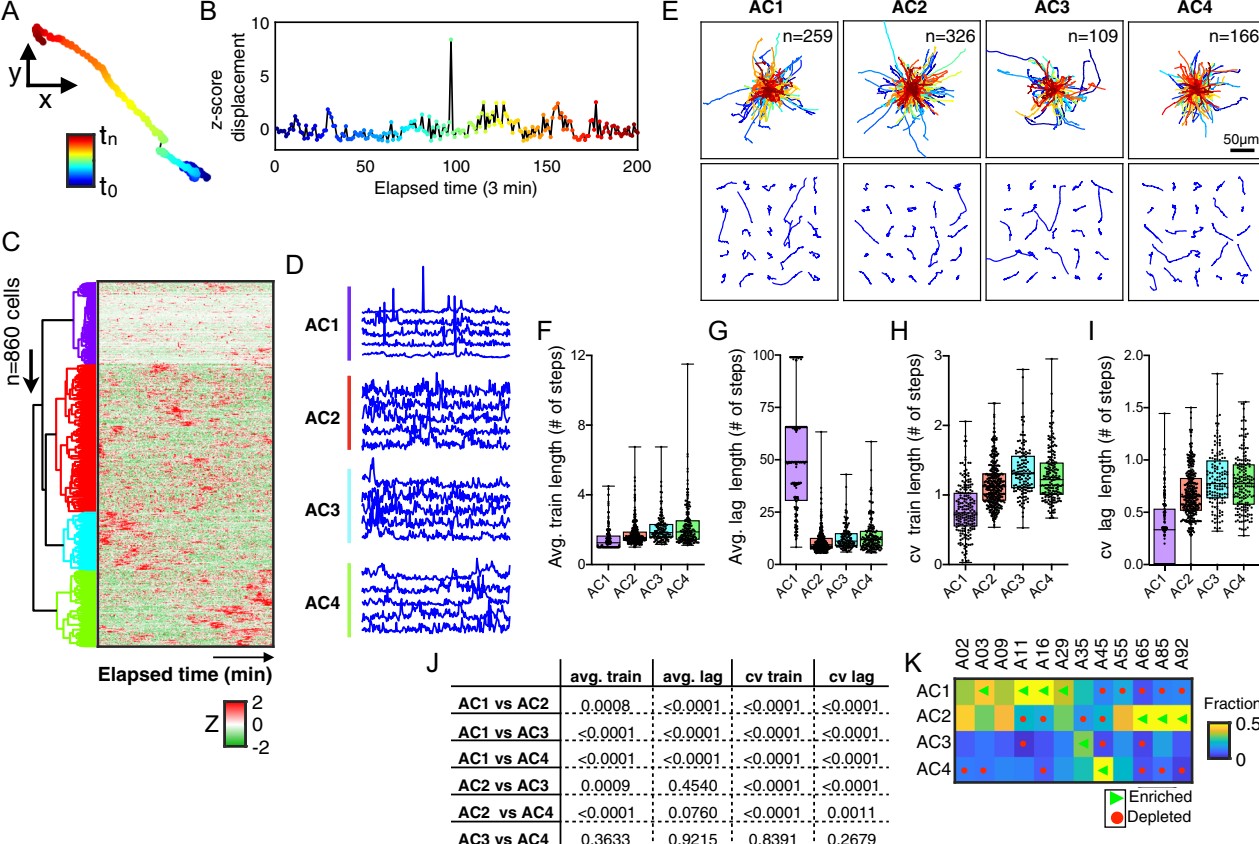

**Fig. 3 Cellular activity describes age-associated motility patterns and heterogeneity. A** Color-coded trajectory of a representative cell as a function of elapsed time (navy blue-to-maroon). **B** Plot showing the corresponding one-dimensional displacement for the same individual cell, here after referred as the activity profile. **C** Heatmap showing the cellular activity profiles for all cells, across all ages (n = 860 cells). Each row represents a single cell and each column from left to right represents elapsed time. Five colored dendrogram branches represent hierarchical clustering using the city block distances with ward linkages along the axes of maximum variation. **D** Line plots showing five representative cellular activity profiles for single cells within each activity cluster. **E** Trajectories of cells within each activity cluster; top panel shows a grid of 25 randomly selected cell trajectories per cluster; bottom panel shows the origin-centered trajectories for all cells per cluster, with the number of cells within each cluster indicated in the upper left corner. **F–I** Box plots showing the extent of bursty dynamics per activity clusters; average train lengths (**F**), average lag lengths (**G**), CV of train lengths (**H**), CV of lag lengths (**I**). Error bars denote the maximum and minimum, with individual data points shown as black dots. **J** Table of p-values showing comparisons among the four activity clusters for average train length, average lag length, cv train length, and cv lag length. **K** Heatmap showing the abundance of cells within each activity cluster per age, green triangles denote significantly enriched and red circles denote significantly depleted (p-value < 0.05).

"City block" distances along the axis of maximum variation ("ward" linkages) to segregate individual cells into activity clusters (ACn). This analysis yielded four activity clusters (Fig. 3C), each being defined based on the frequency and magnitude of the peaks per elapsed time (Fig. 3D). For instance, activity cluster 1 (AC1) described cells having long periods of consistent movements (flat sections of the profiles) with infrequent short-duration bursts, whereas cluster 2 (AC2) exhibited frequent bursts of varying durations.

To visually determine what these activity patterns represent, we plotted the $x-y$ trajectories for cells categorized within each cluster (Fig. 3E). Visual inspection of categorized cells per activity cluster did not reveal overt patterns of movement based on displacement magnitude, however, clusters were populated by a mixture of both motile (i.e., high displacing) and non-motile (i.e., low displacing) cells. These findings suggest that both motile and non-motile cells can exhibit similar activity profiles and consistency in movement/rest relative to their baseline. To further build on this finding, we took the activity profiles of each cell and asked whether applying a point-process analysis (see "Methods" section) could better reveal a biological meaning. Taking the normalized activity profile for each cell, we used a threshold of one standard deviation above the baseline and computed the amount and frequency of movements, thereby defining trains and lags, computed based on the number of consecutive time steps above and below the standard deviation, respectively (Supplementary Fig. 7A, B). Compiling the binarized activity profiles for each cell across all ages (Supplementary Fig. 7C), we computed the distribution of trains (having a binarized activity of "1") and lags (having a binarized activity of "0") (Supplementary Fig. 7D, E). Results indicated that the four clusters can be defined based on the magnitudes of the trains and lags, for instance cells in cluster AC1 displayed significantly shorter trains and long lags (Fig. 3F, G), compared to longer trains and short lags observed for cells in clusters AC3 and AC4. In addition, cells classified as AC1 were more similar in the duration of their trains and lags relative to cells classified in AC3 and AC4, shown by the lower coefficient of variance (Fig. 3H, I). Furthermore, cells from young donors were significantly enriched for AC1 (i.e., suggesting that they move consistently), while cells from older adults were significantly enriched for AC2 with significant depletions for AC1 and AC4 (Fig. 3J and Supplementary Data 5).

Together, the findings indicate that quantifying the bursty dynamics of cell movements relative to their baseline (i.e., activity) provides a complimentary framework to describe age-associated movement behaviors at the single-cell level.

**Cellular heterogeneity, a key feature in describing age-dependent cell motility states.** Given the complimentary information provided by the spatial and activity patterns, we asked whether enrichments or depletions in particular clusters could define age-dependent motility states. By combining both types of information per cell, we could therefore investigate how spatial and activity patterns describe the landscape of age-associated motility phenotypes. Compiling the number of cells belonging to each of the thirty-two possible motility states (i.e., eight spatial clusters and four activity clusters), we plotted frequency heatmaps per age (Supplementary Fig. 8) and age group (Fig. 4A), which revealed topographic regions of high-frequencies and low-frequencies. To determine whether these high-frequency and low-frequency states were significantly enriched or depleted for cells as a function of age, we computed the statistical significance based on the null hypothesis describing the expectation at random. Utilizing a randomization test with 10,000 permutations, we

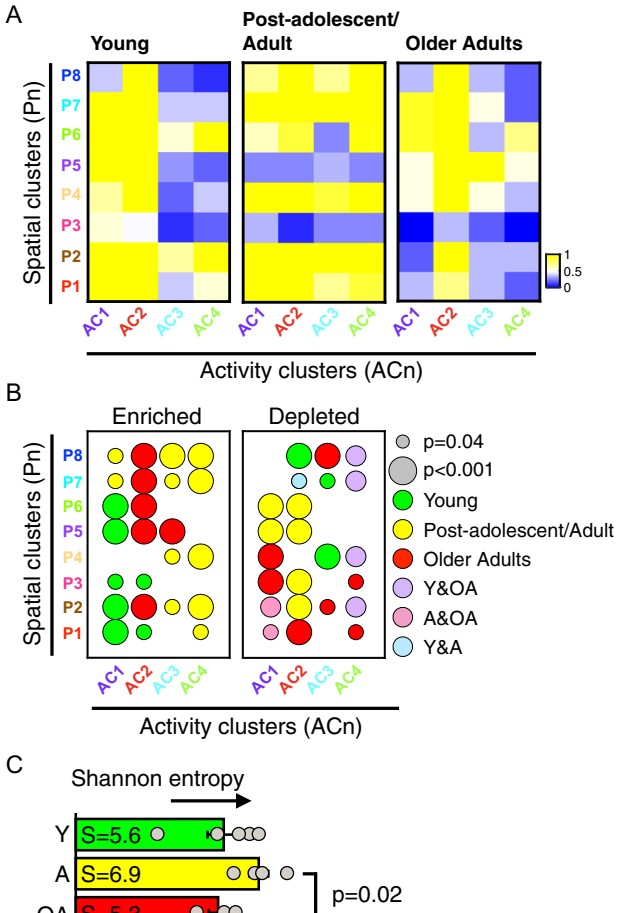

**Fig. 4 Spatial and activity clusters define age-dependent cell motility states. A** Heatmaps showing the frequency per motility state defined by the eight spatial and four activity clusters: young (A02, A03, A09, A11, and A16), post-adolescent/adult (A29, A35, A45, and A55), and older adults (A65, A85, and A92). Color scales indicate the frequencies of cells within each motility state, yellow denotes high frequencies and blue denotes low frequencies. **B** Enrichment and depletion maps corresponding to motility states as a function of age-group. Size of the node denotes the magnitude of the p-value, and colors denote young (green), post-adolescent/adults (yellow), older adults (red). **C** Quantification of the Shannon entropy based on age-group defined motility landscapes.

asked whether the observed frequencies per state was significantly different from the expectation at random for each age group (Supplementary Data 6–7). These results validated what was visualized based on the heatmaps (Fig. 4A), indicating significant enrichments and depletions, respectively. For instance, young donors were significantly enriched ($p < 0.05$) for P1–3 and AC1 and depleted for AC3 and P4, indicating an enrichment for cells exhibiting high displacements, persistence and consistent movement (with few regions of sporadic movement), (Fig. 4B and Supplementary Data 7). Prompted by these differences in the abundances of cells per motility state, we computed the Shannon entropy of each of the three age groups, which showed a statistically significant reduction in cellular heterogeneity for older adults (>65 years) (Fig. 4C).

Together, our result indicate that age-associated cell motility states can be described based on the combined spatial and activity patterns of cellular movements, and is characterized by a redistribution among motility states and reduced heterogeneity for older adults.

## Discussion

In most cell motility analyses, although cells are tracked at the single-cell level for motility experiments, parameters are routinely reported as bulk averages such as average displacements, speed, and persistence. Taking advantage of the single-cell nature of our motility measurements, we quantified changes in cell motility patterns not fully appreciated from standard bulk analysis (Fig. 1A–F). Previous studies using a single-cell approach showed that mouse embryonic fibroblasts and myoblasts display heterogeneous phenotypic states across multiple scales with the potential for cell state-transitions[18,20]. Taking a similar approach, we pooled the individual trajectories for all cells across all ages to identify eight spatial clusters (Fig. 2A–C and Supplementary Fig. 4A) and four activity clusters (Fig. 3C–E). These clusters describe groups of cells having similar motility phenotypes. Combining the spatial and activity patterns of single cells, we defined age-dependent motility states (Fig. 4A, B). This approach and findings highlight the immense amount of information that can be extracted at the single-cell level[5,19,30], and demonstrates that dermal fibroblasts derived from healthy donors comprise a mixture of motile and non-motile cells, which redistributes as a function of age (young, post-adolescent/adults, or older adults). This is an important finding since it provides a different outlook of ageing at the cellular level, and a potential mechanism of how populations of cells encode and manifest age-dependent phenotypes. For instance, our data suggests that this age-associated decrease in overall motility is partly encoded in the cellular heterogeneity, with motility parameters being approximated as a weighted-average based on the abundance of cells within spatially defined patterns of movement (Supplementary Fig. 6A, B). In addition, assessing cellular heterogeneity is an important feature to improve our understanding of emergent phenotypes in the context of ageing in health and disease[31–34].

Developing portraits of ageing at the single-cell level could allow the investigation of novel questions regarding possible age-dependent phenotypic transitions. For instance, we wondered whether we could use the likely progression order among spatial clusters to identify age-associated motility tendencies (see "Methods" section), such as whether the age-associated decrease in cellular persistence preluded the decrease in displacement. To address this, we computed the magnitude of the cross-correlation coefficients among each clusters (P1–P8), and the strength of the correlation based on the abundance of cells within each spatial cluster with age (Supplementary Fig. 9A–C). Together, these correlation trends describe the likely transition order among clusters as a function of age, thereby providing insights into the order of changes in persistence and displacements. Our data suggests that cells tend to decrease their displacements before losing their ability to move in a persistent manner with increasing age (Supplementary Fig. 4D).

In summary, we have demonstrated that singe-cell approaches can capture age-associated emergent patterns of cell motility. We anticipate that the increased implementation of modern single-cell analyses and approaches could lead to a more comprehensive understanding of ageing, with the potential to identify cellular states and phenotypic patterns that could have applications in the development of proxies of ageing in the context of health and disease. While these findings improve our present understanding of cellular determinants of ageing with regard to motility patterns, it remains unclear whether and how cells transition across motility states, and the respective timescales and rates as a function of increasing age. Moreover, the diversity in cellular phenotypes observed with age is likely linked to underlying molecular programs, cellular subtypes and cell cycle states that altogether influence the motility patterns of cells[5,35]. We anticipate that future work is needed to address this, which will require the development of new technologies and imaging modalities to dynamically assess both motility and the underlying molecular status of cells (e.g., cell cycle, epigenetic status—DNA and histone methylation, protein expression and localization) in a large cohort of well annotated healthy donors (cross-sectional and longitudinal), imaged for prolonged periods of time (order of days).

## Methods

**Cell culture**. A panel of twelve early-passage, primary dermal fibroblasts ranging in age from 2 to 92 years old (GM00969, GM05565, GM00038, GM00323, GM06111, AG04054, AG11796, AG08904, AG09162, AG12940, AG09558, and AG09602), were obtained from Coriell Biobank cell repository (Camden, NY, USA), from collections comprising the Baltimore Longitudinal Study of Aging (BLSA) and the NIGMS apparently healthy controls. Cells were cultured in high-glucose (4.5 mg/ml) DMEM (Gibco), supplemented with 15% (vol/vol) fetal bovine serum (Hyclone), and 1% (vol/vol) penicillin–streptomycin (Gibco). Cell cultures were passaged every three to four days or when flasks were at ~80% confluence, for a maximum of five passages used for motility experiments. Data for cell trajectories for 860 cells across all ages can be found in Supplementary Data 2. For motility experiments cells were seeded on type-1 collagen-coated substrates in order to maintain viable, adherent and active fibroblasts, (i.e., akin to other tissue culture coatings, e.g., poly-L-lysine). In addition, since fibroblasts secrete large amounts of collagens, we rationalized that having a type-1 collagen-coating (the most abundant in skin) would allow the cells to move more efficiently.

**Quantification of cell motility**. Fibroblasts were seeded at low density (~2000 cells/ml) onto type-1 collagen-coated (50μg/ml) substrates in 24-well plates and allowed to adhere overnight. Once cells attached, the plate was mounted unto a Nikon TE2000 microscope equipped with a motorized stage with X–Y–Z controls (Prior scientific) and environment control to maintain physiological conditions of temperature (37 °C), Carbon dioxide (5%) and humidity (Pathology Devices). Phase-contrast images were recorded every 3 min for 16 h using a Cascade 1K CCD camera (Roper Scientific) with a low magnification 10× Plan Fluor objective (numerical aperture, 0.3; Nikon). Cell motility parameters were determined from x–y coordinates obtained from the cell-centroid tracking of individual cells (MetaMorph). Cells undergoing division, moved out of frame, or had long contact times with other cells were omitted from analysis. Only cells having ten continuous hours of trackable movements that did not meet the above exclusion criteria were used for the final analysis. X–Y coordinates were then exported for analysis in Matlab for the quantification of spatial[24] and activity parameters described below.

To quantify the spatio-temporal patterns of motility, we analyzed bulk cell movements based on trajectories using the Anisotropic Persistent Random Walk model (APRW)[24]. From this analysis we generated the parameters that describe the movements of the cells, (MSD60, MSD6, SP60, SP6, Pp, Pnp, Dp, Dnp, Dtot, and φ), together with the mean-squared displacements, the auto-correlation function of velocities and the angular velocity magnitudes, which were computed and fitted based on the following equations:

$$\mathrm{MSD}(\tau) = \left\langle x(t+\tau) - x(t))^2 + y(t+\tau) - y(t))^2 \right\rangle,$$

$$\mathrm{MSD}_{\mathrm{p}}(\tau) = S_{\mathrm{p}}^2 P_{\mathrm{p}}\left(\tau - P_{\mathrm{p}}\left(1 - e^{-\tau/P_{\mathrm{p}}}\right) + 2\sigma_{\mathrm{p}}^2\right),$$

$$\mathrm{MSD}_{\mathrm{np}}(\tau) = S_{\mathrm{np}}^2 P_{\mathrm{np}}\left(\tau - P_{\mathrm{np}}\left(1 - e^{-\tau/P_{\mathrm{np}}}\right) + 2\sigma_{\mathrm{np}}^2\right),$$

where $S$ is the cell speed, $P$ is the persistence time, $2\sigma^2$ is the noise (error) in the position of the cell, $\tau = n\Delta t$ and $n = 1, 2, \ldots N_{\min}-1$, $\Delta t$ is the size of the time step:

$$\mathrm{ACF}(\tau) = \left\langle \mathrm{d}x(t)\mathrm{d}x(t+\tau) + \mathrm{d}y(t)\mathrm{d}x(t+\tau)\right\rangle,$$

where

$$\mathrm{d}x(t) = x(t + \mathrm{d}t) - x(t),$$

$$\mathrm{d}y(t) = y(t + \mathrm{d}t) - y(t),$$

$$\tau = n\mathrm{d}t,$$

$$n = 1, 2, \ldots .$$

**Defining spatial motility clusters**. Cell motility parameters describing the cells' displacements, speeds, persistence times, diffusivities, and the spatial persistence/anisotropy were computed using the anisotropic persistence random walk model (APRW)[4,5]. For bulk motility analysis, ten motility parameters (MSD60, MSD6, SP60, SP6, Pp, Pnp, Dp, Dnp, Dtot, and φ) were computed for each cell. For bulk analysis, parameters were averaged across all cells per age and the magnitude of the Pearson correlation coefficient was determined (Fig. 1F and Supplementary Fig. 1). For single-cell analyses, the distributions of motility parameters were log normalized to generate a normal distribution per parameter (Supplementary Fig. 2). This resulted in the reduction of motility parameters from ten to eight (MSD60,

MSD6, Pp, Pnp, Dp, Dnp, Dtot, and ϕ), since the normalized MSD6 was roughly equal to SP6, and MSD60 equal to SP60, (i.e., $SP_x = \frac{\sqrt{MSD_x}}{\tau_x}$, where $\tau$ = time lag). Following the normalization, the eight motility parameters were used to define the spatial clusters (Pn). Spatial clusters were defined based on the abundance of cells having similar magnitudes of the eight features defined in the APRW model. Specifically, the motility features were computed for each cell and compiled for all cells across all ages. Importing this data into Matlab, we performed unsupervised hierarchical clustering analysis based on the Cityblock distances along the axis of maximum variation (ward linkages). This clustering analysis resulted in the stratification of cells into eight clusters (P1–P8).

**Determining the likely progression order**. To determine the likely progression order with age, we computed the magnitude of the correlation for the abundance of cells per cluster with age, and the cross correlation among clusters. Once the correlation coefficients were determined, the correlations of cell abundances with age were ranked to determine the overall progression order. In addition to determine the linkages (length—strength of cross-correlation) of clusters we also ranked the correlations among clusters to determine the cluster-to-cluster proximity. Once both sets of correlations were compiled, a network were constructed such that the size of the nodes were scaled based on the number of cells in each cluster, and the cluster-cluster proximity denoted the strength of the cross correlation. The magnitudes of the Pearson correlation coefficients were computed in Matlab, ($R$ = corrcoef(A), where "$R$" is the Pearson correlation coefficient and "$A$" is the data matrix of cell abundances.

**Quantifying the activity profiles per single cell**. To determine the activity profiles of individual cells, the raw $x-y$ trajectories for each cell were converted into 1-dimensional displacement trajectories (Fig. 3A, B). Here, the temporal displacement frequencies and the presence/absence of spikes (burst of movement) and trains (continuous bursts of movements) defines the activity space. We computed the activity based on the displacements according to this equation:

$$x(t) = \sqrt{\Delta x^2 + \Delta y^2},$$

where $\Delta x = x(t) - x(t-1)$ and $\Delta y = y(t) - y(t-1)$, are the changes in the vector components of the cell movements in Cartesian coordinates at time $t$, $\Delta t$ is the time step between different measurements of cell positions.

Once this was computed for each cell, the magnitudes of the displacements were $z$-score normalized per unit time so that each cell was normalized to its own baseline movements, and comparable on the same numerical scale.

$$z = \frac{x - \mu}{\sigma},$$

where $z$ is the $z$-score, $x$ is the magnitude of the variable, $\mu$ and $\sigma$ denote the mean values and the standard deviation, respectively.

The activity was computed for all 860 single cell across all ages, then we performed unsupervised hierarchical clustering analysis to delineate groupings of cells having similar activity profiles. This was done in Matlab based on the Cityblock distances along the axis of maximum variation (ward), which yielded five clusters that we later interrogated to identify cluster-dependent motility patterns.

To calculate the binarized activity, the continuous activity profile was transformed into a binary matrix of 1's and 0's denoting trains and lags. Trains denote bursts of movements one standard deviation above the baseline movement, and lags denotes time steps at baseline and below the threshold. Distribution of trains and lags were computed by compiling the series of 1's and 0's within each temporal activity pattern for all cells and compiles per activity cluster, respectively.

**Computing cellular heterogeneity**. To quantify the heterogeneity among cells (both for spatial and activity clusters) we utilized the Shannon entropy. Here the entropy $S$ was calculated as:

$$S = -\sum_i^N p_i \cdot \ln(p_i),$$

where $p_i$ corresponds to the fractional abundance of cells within the particular cluster, for each age group $N$ (young, middle age, and older adults). The entropy was determined on both a per age and age-groups, based on the abundance of cells per cluster (i.e., both spatial and activity). Entropy per age group was computed by taking the average of entropies per age (young-A02, A03, A09, A11, and A16; post-adolescent/adults-A29, A35, A45, and A55; older adults-A65, A85, and A92).

**Quantifying enrichments and depletions of motility clusters and states**. To determine whether the clusters defined above (spatial and activity) were significantly depleted or enriched as a function of age or age groups, we utilize a randomization strategy. For each cluster "i", we obtain two $p$-values per age corresponding to null hypotheses of under-representation (depletion) and over-representation (enrichment), respectively. Specifically, for each cluster "i" and for each age or age-group "k", we compare the observed frequency of cells with a distribution of expected frequencies of cells of age $k$, built by from $N$ samples

(50,000 permutations) of the size of cluster "i".

$$P_e = \frac{\exp \leq \text{obs}}{N},$$

$$P_d = \frac{\exp \geq \text{obs}}{N},$$

where "exp" denotes the simulated abundance based on the null hypothesis, and "obs" denoted the observed abundance of cells per cluster/state. $P_e$ and $P_d$ denote the $p$-values for the enrichment and depletion, respectively.

**Statistics and reproducibility**. All experiments were conducted with in-plate technical controls in triplicates. The specific number of cells used in analysis are denoted in the main figures for each section. Correlation analysis was conducted using the Pearson correlation coefficients, and statistical significance were assessed using either $t$-tests or one-way ANOVA. To compute the significant enrichments and depletions within each of the 40 defined motility states, we employed a randomization enrichment test for 50,000 permutations and compared to observed frequencies. Significance was determined based on the magnitude of the $p$-values. All cells measured was used for analysis, no samples were intentionally removed from the dataset.

**Reporting summary**. Further information on research design is available in the Nature Research Reporting Summary linked to this article.

## Data availability

The authors declare that all data supporting the findings in this study are available within the paper and its Supplementary documents/information.

## Code availability

Detailed descriptions of our approach and code utilized is either provided in the supplementary documentation or is already available through other published literature.

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

## Acknowledgements

We acknowledge the financial support for this study from the National Institutes of Health, U01AG060903 (D.W., J.W., J.M.P., and P.W.), U54CA143868 (D.W.), R01CA174388 (D.W.), P30AG021334 (J.W.). Funds to support this study were provided by the Johns Hopkins University Older Americans Independence Center of the National Institute on Aging (NIA) under award number P30AG021334 (JMP).

## Author contributions

J.M.P. and D.W. conceived study design and analysis; J.M.P., M.P.P., J.D., S.M.G., W.W., and K.T. performed experiments and collected data; J.M.P., N.Z., and D.W. conceived analysis and analyzed data; J.M.P., D.W., N.Z., J.W., and H.J. interpreted results; D.W., J.W., and P.W. supervised study; J.M.P. and D.W. wrote manuscript; all authors contributed to reviewing and editing the manuscript.

## Competing interests

The authors declare no competing interests.
