## [Peer Review File · Communications Biology]

Reviewers' comments:

Reviewer #1 (Remarks to the Author):

This study by Phillip et al., utilizes a single cell gauging of a plethora of motility configurations in fibroblastic cells harvested from human skin of healthy, 2- to 92-year-old, donors. The study proposes that single cell motility patterns change with age because aging individuals show decreased fibroblastic heterogeneity as well as dampened motility dynamics (i.e., train/lag lengths). The study also proposes that motility parameters, representative of this narrowing in cellular heterogeneity, could serve as phenotypic aging biomarkers. An apparent predicament associated with this study is that the same team published a similar study, reaching related conclusions, in 2017, questioning the level of novelty. Further, the practicality of utilizing single cell motility information as an "aging" biomarker is not trivial and the biomarker utility was not validated/confirmed. Some added concerns, which if addressed might increase the potential impact of the study, are listed below.

It is apparent that many of the measured parameters are similar between the 2017 and the current study (and maybe even some of the cells used are the same). Hence, it is important that the authors clearly highlight the differences and similarities between the two studies, underscoring the novelty aspects introduced by the current one.

Despite being highly informative, the study is mostly observational.

It is possible that, by using collagen I coated surfaces, the authors unintentionally selected a subgroup of motility-related traits. Thus, it is possible that additional traits were not apparent (i.e., limited by the flat and singular protein substrate utilized). Further, homeostatic/quiescent fibroblasts are known to drastically alter their behavioral traits, especially motility, in the presence of fibrin clots (generated by leaky vessels when homeostatic equilibrium is altered during development, wound healing and chronic diseases <https://pubmed.ncbi.nlm.nih.gov/8424460/>). Also, collagenous matrices (as opposed to single protein coating) are made by these cells much later upon resolving a wound in vivo. Then again, during development, mesenchymal-like fibroblastic cells present with motility patterns mostly within fibronectin-rich extracellular matrices. Hence the flat coating collagen I selection needs justification by the authors. This point does not take of the fact that the assay presented, as suggested by the authors, could serve as an age-related functional heterogeneity (or motile fibroblastic homogeneity) biomarker.

Some statements are rather overreaching. For example, without providing a confirmation cohort, the results describing reduced heterogeneity and dynamic changes between train and lag lengths (as well as others), cannot be interpreted as bona fide "age biomarkers." It is recommended that the authors test the validity of the claimed biomarker in a blinded validation cohort, composed of skin fibroblasts collected from very young, vs very old individuals, and test whether results stratify with the ones from the discovery cohort presented in the initial submission of this work.

The following point constitutes a suggestion to the authors that could improve the potential impact, yet will make no difference to the central point being conveyed. The combined list of parameters depicted in this study seems to be very useful. Hence, as part of the supplemental materials, perhaps the authors could consider providing a formatted set (i.e., as a spreadsheet) in which users could input their single cell measured data (i.e., x/y coordinates per time for each cell) to obtain numerical and graphical outputs representative of each of the parameters offered by the study. This, could serve the broad cell biology community and might progress into an "Image plugin;" single cell motility/ cell heterogeneity analysis tool.

Supplemental figure3A may merit a main figure panel, as opposed to supplementary figure panel. Then again, if this is similar to the 2017 data, then it belongs on the supplementary materials as presented.

Could Figure 2D be expanded as a Supplemental Figure in which each of the age samples is shown individually as a single colored set overlaid onto the same 8 clusters? This way the distribution shift (from P1-3 to P6-8 associated with age) could be a bit clearer. Figure 2E will then complement better these data.

Reviewer #2 (Remarks to the Author):

The manuscript entitled, "Fractional re-distribution among cell motility states during ageing" from Phillip et al uses statistical analyses of cell mean squared displacements to classify dermal fibroblasts from 12 healthy donors spanning the ages of 2-92. For each cell line 30-105 cells are tracked over at least 10 hours (for total of 860 tracked cells) and the data is analyzed using the anisotropic persistence random walk model, which was previously described in their 2015 Nature Protocol paper. Multiple single cell motility parameters are derived using this approach, including parameters related to displacement magnitude (total diffusivity, persistent diffusivity, and mean square displacements for early and late time scales), migratory persistence (persistence in primary axis and anisotropy), and orthogonal migration (persistence and diffusivity in non-primary axis). These motility parameters were quantified for the 860 individual cells. Hierarchical clustering was then used to identify cell-based and parameter-based groupings ('city block' distances) and variation in these groupings ('ward linkages') and how they vary with cell donor age. The statistical approach used in analyzing this data is extremely thorough and leads to some important discoveries. The individual traces are also analyzed for burst motion to identify periods of motion (trains-given 1's) and lags (where cells are immobile-assigned 0's). This analysis is used to examine the effects of aging on migratory behavior and consistency. They conclude that heterogeneity in cell motility is reduced progressively with aging. This reduced cellular heterogeneity has been shown for senescent vs. pre-senescent cells, but this paper clearly shows how the heterogeneity is reduced with age. Overall, these are very interesting findings. Some concerns are listed below.

1. How does the variation in the number of cell traces (30-105) affect the statistical analyses used in this study (including groupings and bin sizes)?
2. For sample A11, Figure 1E shows variation in reported z-scores and the traces in G are more heterogeneous than other cells (like A16, A9), not clear if tracing more cells might resolve this inconsistency.
3. Parameter groupings appeared to combine similar parameters (e.g., MSD and Speed). These parameters also appear to have similar z-scores in most cases (Fig. 1E-F); however, MSD6 and S6 often differ and Pnp and Dnp, especially for A55 and A85. What contributes to these differences?
4. Discussion of the spatial and activity clustering was somewhat vague.

Response to reviewers:

We thank the reviewers for the critical review of our manuscript, and providing comments and suggestions that improved the quality and clarity of our work. Below, we have included a point-by-point response to all the comments raised, all the additions/modifications to the manuscript text is noted in 'blue'.

Reviewers' comments:

Reviewer #1 (Remarks to the Author):

This study by Phillip et al., utilizes a single cell gauging of a plethora of motility configurations in fibroblastic cells harvested from human skin of healthy, 2- to 92-year-old, donors. The study proposes that single cell motility patterns change with age because aging individuals show decreased fibroblastic heterogeneity as well as dampened motility dynamics (i.e., train/lag lengths). The study also proposes that motility parameters, representative of this narrowing in cellular heterogeneity, could serve as phenotypic aging biomarkers. An apparent predicament associated with this study is that the same team published a similar study, reaching related conclusions, in 2017, questioning the level of novelty. Further, the practicality of utilizing single cell motility information as an "aging" biomarker is not trivial and the biomarker utility was not validated/confirmed. Some added concerns, which if addressed might increase the potential impact of the study, are listed below.

1. It is apparent that many of the measured parameters are similar between the 2017 and the current study (and maybe even some of the cells used are the same). Hence, it is important that the authors clearly highlight the differences and similarities between the two studies, underscoring the novelty aspects introduced by the current one.

Although we are building on the findings from our 2017 Nature BME paper, we are presenting findings and using analysis/computational approaches that are new for this study.

In our 2017 study, a key goal was to evaluate whether we can use biophysical properties of cells to determine the cellular age, and whether these measures can be used as biomarkers of ageing. In that study we evaluated 5 main classes of biophysical measurements, including morphology, motility and cell mechanics, with a greater focus on morphology. In this study we are digging deeper into the dynamics of cells, i.e. cell motility. For the motility analysis in 2017 study, we did utilize APRW analysis to compute motility parameters, however, we stuck to the traditional analyses in using aggregate values of cell motility to identify global patterns.

In the present study, we are going beyond traditional approaches to determine what additional information can be learnt from taking advantage of the single cell nature of the data. Two key advances of our new approach is that it allows the identification of age-associated spatial and activity clusters that better define motility patterns and provides a straightforward way to compute the degree of cell-to-cell variations as a function of age, (demonstrating that there is additional information that we can learn by looking beyond aggregate data). In addition, it brings forth a new way to assess motility patterns by taking advantage of the single cell information, and the importance of heterogeneity.

We have also added text at various points throughout the manuscript for clarity in this regard.

2. Despite being highly informative, the study is mostly observational.

We agree that the study is observational, and we do not provide a mechanism as to the molecular drivers for the observed age-associated phenotypes described (i.e. different spatial and activity clusters). Although we note these limitations in our discussion, a key goal for this study is to determine whether quantifying single-cell motility patterns provide additional information about the motility of ageing cells that is not apparent from the aggregate of the same data.

As such, we have employed robust computational strategies and have used statistical metrics (such as null hypothesis testing, randomized permutations and different clustering methods), most of which are not traditionally applied to motility data. Furthermore, we have demonstrated the utility of this approach and present interesting findings for ageing cells. While we are interested in identifying some of the underlying molecular drivers of the observed phenotypes, it is beyond the scope of the current study, as it will require the development of novel experimental approaches and tools to properly tease apart these underlying factors. We are currently work on developing such tools and will present finding in subsequent studies. However, even with these limitations regarding molecular mechanisms, our study provides a step toward a better understanding of single-cell motility and age-associated motility patterns. Furthermore, the approach presented here is applicable to all types of motility data and not just ageing data. As a result, this approach could be beneficial to the field at large in terms of how one thinks and quantifies cell motility.

3. It is possible that, by using collagen I coated surfaces, the authors unintentionally selected a subgroup of motility-related traits. Thus, it is possible that additional traits were not apparent (i.e., limited by the flat and singular protein substrate utilized). Further, homeostatic/quiescent fibroblasts are known to drastically alter their behavioral traits, especially motility, in the presence of fibrin clots (generated by leaky vessels when homeostatic equilibrium is altered during development, wound healing and chronic diseases <https://pubmed.ncbi.nlm.nih.gov/8424460/>). Also, collagenous matrices (as opposed to single protein coating) are made by these cells much later upon resolving a wound in vivo. Then again, during development, mesenchymal-like fibroblastic cells present with motility patterns mostly within fibronectin-rich extracellular matrices. Hence the flat coating collagen I selection needs justification by the authors. This point does not take of the fact that the assay presented, as suggested by the authors, could serve as an age-related functional heterogeneity (or motile fibroblastic homogeneity) biomarker.

We agree that the selection of experimental conditions is critical, and could impact and/or limit the use of the findings due to either bias or the use of non-physiological contexts. Since the focus of this study was not to identify the molecular drivers for the differences in motility with age, we selected the experimental conditions not trying to match the physiological context exactly, but on using a substrate (i.e. collagen-1 coating) containing an appropriate ECM protein (collagen is abundant in the skin) that was able to keep the cells viable, adherent and active. While we do acknowledge that the cells are taken out of physiological context (e.g. 2D cultures, etc.), the fact that we are still able to identify age-associated trends speaks to the intended utility. In this case, the collagen-coating should be thought of as a tissue culture tool, akin to poly-l-lysine that is used to coat surfaces to boost cell adhesion. In addition, because fibroblasts secrete collagens, we rationalized that having this collagen coating will allow them to move more efficiently for the short durations of the experiment.

We have included additional information in the materials and methods section to add clarity.

4. Some statements are rather overreaching. For example, without providing a confirmation cohort, the results describing reduced heterogeneity and dynamic changes between train and lag lengths (as well as others), cannot be interpreted as bona fide “age biomarkers.” It is recommended that the authors test the validity of the claimed biomarker in a blinded validation cohort, composed of skin fibroblasts collected from very young, vs very old individuals, and test whether results stratify with the ones from the discovery cohort presented in the initial submission of this work.

We apologize for the overreaching statements; we have revised and modified the text within the main manuscript to tone down some of the claims. The reviewer is correct that we have not validated some of these features (i.e. trains and lags) as bona fide ageing biomarkers, however, because of their age-associations we are suggesting that they contain ageing information, and merit further study and analysis.

The following point constitutes a suggestion to the authors that could improve the potential impact, yet will make no difference to the central point being conveyed.

5. The combined list of parameters depicted in this study seems to be very useful. Hence, as part of the supplemental materials, perhaps the authors could consider providing a formatted set (i.e., as a spreadsheet) in which users could input their single cell measured data (i.e., x/y coordinates per time for each cell) to obtain numerical and graphical outputs representative of each of the parameters offered by the study. This, could serve the broad cell biology community and might progress into an “Image plugin;” single cell motility/ cell heterogeneity analysis tool.

Thank you for this suggestion, we agree that having an open tool where interested users can input their data for analysis could potentially impact the broader cell biology community. We are in the process of developing such a method that we plan to release in a subsequent study. In the meantime, for this current study, we have provided our full dataset containing the coordinates for all the cells tracked and analyzed, together with a detailed description of how the data was processed. That way readers can replicate our findings and apply to their own datasets.

6. Supplemental figure 3A may merit a main figure panel, as opposed to supplementary figure panel. Then again, if this is similar to the 2017 data, then it belongs on the supplementary materials as presented.

We have moved the plot, previously supplemental figure 3A, to the main figure as Figure 2B. This analysis is entirely new and the plot was not included in our 2017 Nature BME paper.

7. Could Figure 2D be expanded as a Supplemental Figure in which each of the age samples is shown individually as a single colored set overlaid onto the same 8 clusters? This way the distribution shift (from P1-3 to P6-8 associated with age) could be a bit clearer. Figure 2E will then complement better these data.

We agree with the reviewer that expanding Figure 2D will add clarity, we have now included this panel as supplementary Figure 5.

Reviewer #2 (Remarks to the Author):

The manuscript entitled, "Fractional re-distribution among cell motility states during ageing" from Phillip et al uses statistical analyses of cell mean squared displacements to classify dermal fibroblasts from 12 healthy donors spanning the ages of 2-92. For each cell line 30-105 cells are tracked over at least 10 hours (for total of 860 tracked cells) and the data is analyzed using the anisotropic persistence random walk model, which was previously described in their 2015 Nature Protocol paper. Multiple single cell motility parameters are derived using this approach, including parameters related to displacement magnitude (total diffusivity, persistent diffusivity, and mean square displacements for early and late time scales), migratory persistence (persistence in primary axis and anisotropy), and orthogonal migration (persistence and diffusivity in non-primary axis). These motility parameters were quantified for the 860 individual cells. Hierarchical clustering was then used to identify cell-based and parameter-based groupings ('city block' distances) and variation in these groupings ('ward linkages') and how they vary with cell donor age. The statistical approach used in analyzing this data is extremely thorough and leads to some important discoveries. The individual traces are also analyzed for burst motion to identify periods of motion (trains-given 1's) and lags (where cells are immobile-assigned 0's). This analysis is used to examine the effects of aging on migratory behavior and consistency. They conclude that heterogeneity in cell motility is reduced progressively with aging. This reduced cellular heterogeneity has been shown for senescent vs. pre-senescent cells, but this paper clearly shows how the heterogeneity is reduced with age. Overall, these are very interesting findings. Some concerns are listed below.

1. How does the variation in the number of cell traces (30-105) affect the statistical analyses used in this study (including groupings and bin sizes)?

We agree that having a large sample size for each condition is critical to the analysis. In this study we have used 12 donor samples, each having an average of 71 cells per condition (median=68 cells), with sample A11 having the least number of cell (n=30 cells), and sample A29 having the most cells (n=129 cells). For the bulk motility analysis presented in Figure 1, we have confidence in the results as both A11 and A29 follow similar trends to other samples within the age group. Note that the three age groups (young, post-adolescent/adults, older adults) were determined based on cutoffs used in the ageing literature and were not arbitrarily set by the data presented in the paper.

Secondly, in order to further minimize any effect of sample size per condition on the statistics and robustness of the analysis and findings, we combined cells across all conditions to identify and categorize patterns of spontaneous migration. These spatial migration clusters were assessed and confirmed using different clustering and statistical methods including hierarchical clustering, tSNE and null hypothesis testing. Furthermore, by assessing the fractional abundance of motility clusters per age (Figure 2F), we were again able to identify similar trends for samples A11 and A29 to other samples with their age group, respectively.

2. For sample A11, Figure 1E shows variation in reported z-scores and the traces in G are more heterogeneous than other cells (like A16, A9), not clear if tracing more cells might resolve this inconsistency.

We thank that reviewer for the comment. In re-checking the data to provide an adequate response to the comment raised, we realized that the Figure 1E in the manuscript version

submitted was from an earlier iteration of the plot with fewer cells. We have now corrected this and updated the plot. While the update did change the plot slightly, the overall trends and findings remain the same. We have also included the raw data and z-scores as supplementary table S2.

Regarding sample A11 (also see comment above), we agree that having more cells tracked will boost the robustness of the individual result and may homogenize the motility parameters and traces. However, we are limited in our ability to include additional cells at the moment. While the motility parameters show lower values (white-pink vs. green-white) than other samples within the young age group for the aggregate analysis (Figure 1E), we retain the overall age-trend with or without the inclusion of sample A11. In addition, since we are not trying to predict the age of the individual in this study, it is an acceptable error.

Regarding the z scores used in the heatmap (Figure 1E and Supplementary table S2), each parameter was normalized using the mean and standard deviation across ages (i.e. normalized per row, since they have differing numerical scales).

3. Parameter groupings appeared to combine similar parameters (e.g., MSD and Speed).

These parameters also appear to have similar z-scores in most cases (Fig. 1E-F); however, MSD6 and S6 often differ and Pnp and Dnp, especially for A55 and A85. What contributes to these differences?

For Figures 1E-F (we have updated figure 1E see comment above), we show that parameters group based on the type of information that they provide (not all parameters are orthogonal). As the reviewer pointed out in Figure 1E the MSD and Speeds correlate, as the Sp is computed based on the MSD and time lag. Pearson correlation for aggregated MSD6 and S6 is calculated as 0.98, with principal component analysis (now Figure 2B) showing a complete overlap in PCA space. For this reason, we have chosen to use only MSD in figure 2 onwards. Regarding the correlation between Dnp and Pnp, we see a Pearson correlation of 0.70. This is also seen in correlations among different cell types (Wu and Giri et al, 2014). Explanations for these difference can be traced back to how the parameters are computed. The diffusivity in the non-primary axis of migration is computed as $D_{np} = (S_{np}^2 * P_{np}) / 4$, where Pnp is the persistence time in the non-primary axis, and Snp being the fit speed in the non-primary axis based on the APRW, see Wu and Giri et al. PNAS 2014 for more details. Briefly, both values of Pnp and Snp are computed based on fitting the model, therefore we do not expect to see perfect correlation with Dnp and either Snp or Pnp.

4. Discussion of the spatial and activity clustering was somewhat vague.

We have modified the text in the results and discussion to boost the clarity regarding the meaning of the spatial and activity clusters.

REVIEWERS' COMMENTS:

Reviewer #2 (Remarks to the Author):

The authors have addressed all of my concerns.

Reviewer #3 (Remarks to the Author):

I actually don't have that many comments. I feel that the authors in their manuscript and in their response to earlier referee reports have made a clear case that there is significant age-dependence in a variety of motility parameters and that the most likely interpretation of this bulk result is the distribution of single-cell motility phenotypes becomes altered. There is no molecular explanation offered, but I guess the results are useful by themselves; hence this lack of understanding makes the work less exciting but still publishable. Also, there is little quantitative discussion of the benefits of using motility as an age marker as opposed to some already published idea, for example related to epigenomics. This again limits the usefulness of this work, but again should not preclude publication.